# Switchable chemoselective aryne reactions between nucleophiles and pericyclic reaction partners using either 3-methoxybenzyne or 3-silylbenzyne

Hongcheng Tan[1], Shuxin Yu[1], Xiaoling Yuan[1], Liyuan Chen[1], Chunhui Shan[2], Jiarong Shi[1] & Yang Li [1] ✉

Arynes are known to serve as highly reactive benzene-based synthons, which have gained numerous successes in preparing functionalized arenes. Due to the superb electrophilic nature of these fleeting species, however, it is challenging to modulate the designated aryne transformation chemoselectively, when substrates possess multiple competing reaction sites. Here, we showcase our effort to manipulate chemoselective control between two major types of aryne transformations using either 3-methoxybenzyne or 3-silylbenzyne, where nucleophilic addition-triggered reactions and non-polar pericyclic reactions could be differentiated. This orthogonal chemoselective protocol is found to be applicable between various nucleophiles, i.e., imidazole, N-tosylated/N-alkyl aniline, phenol, and alcohol, and an array of pericyclic reaction partners, i.e., furan, cyclopentadiene, pyrrole, cycloheptatrienone, and cyclohexene. Beyond arylation reactions, C–N bond insertion, Truce–Smiles rearrangement, and nucleophilic annulation are appropriate reaction modes as well. Moreover, this chemoselective protocol can find potential synthetic application.

Since its structural determination by Kekulé in 1865[1], benzene has been found in pharmaceuticals, bioactive natural products, agrochemicals, and functional organic molecules. Because of the constant and emerging demands on substituted arenes from fine chemical manufacture to drug development in pharmaceutical companies and to fundamental research in laboratories, strategies that can efficiently access those scaffolds have been actively pursued. As one of the most active organic intermediates, benzyne has long been recognized as a versatile benzene-based building block in college textbooks and in modern advanced synthesis, which could expeditiously assemble numerous polyfunctionalized benzenes as well as benzofused compounds under mild conditions (Fig. 1A)[2–14]. To date, there are three major types of benzyne reactions: nucleophilic addition-triggered reactions, pericyclic reactions, and transition metal-catalyzed reactions. Some notable achievements in the past decade include the preparation of natural products[15–18], polyarylarenes[19], nanographenes[11,20], and carbon nanobelt[21] as well as generation methods[22–25]. Particularly, it has been used as metal-free alternative of transition metal-catalyzed cross-coupling reactions in preparing biaryls[26]. Accompanied with the advances of synthetic methodologies, the groups of Garg and Houk systematically investigated regioselectivity in aryne transformations using distortion-interaction model, where the preferred reaction site on distorted arynes could be predicted with respect to polar arynophiles (Fig. 1A)[27–29].

Despite those fruitful achievements, chemoselectivity as a fundamental facet in aryne chemistry has not been addressed. Because of

[1]School of Chemistry and Chemical Engineering, Chongqing University, 174 Shazheng Street, Chongqing 400030, PR China. [2]College of Chemistry, Chongqing Normal University, Chongqing 401331, PR China. ✉e-mail: y.li@cqu.edu.cn

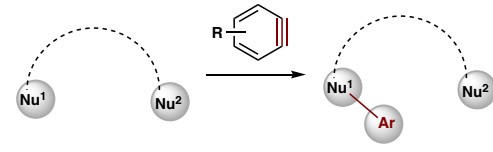

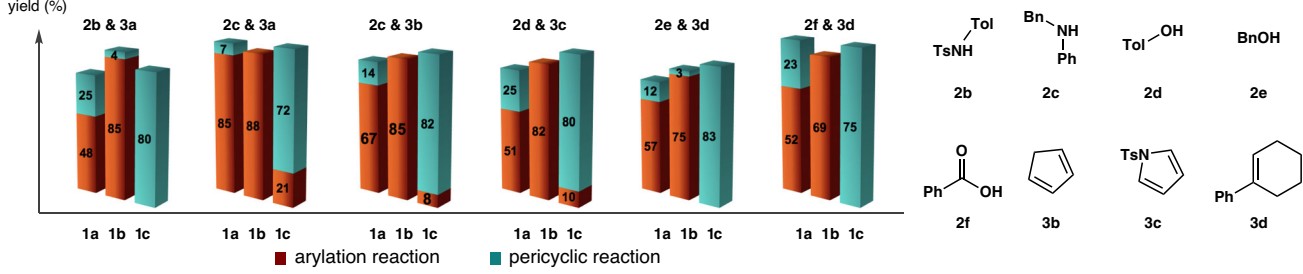

**Fig. 1 | Background and our work. A** Benzyne and distorted arynes with either electron-withdrawing group (EWG) or electron-donating group (EDG) on the 3-position that contain preferred site of nucleophilic addition. **B** Previous chemoselective method employs the inherent reactivity difference between two nucleophiles to realize chemo- and/or site-selectivity, albeit with only limited examples.

**C** Trapping reactions between imidazole and furan: [a]Conditions: aryne precursor **1** (0.5 mmol), imidazole (**2a**) (1.5 mmol), furan (**3a**) (1.5 mmol), CsF (1.5 mmol), and additive (1.0 mmol) in acetonitrile (5 ml) at room temperature overnight. [b]Isolated yield. [c]A mixture of *ortho*- and *meta*-isomers. **D** Selected intermolecular competing reactions. **E** Chemoselectivity study on compound **sub-1**.

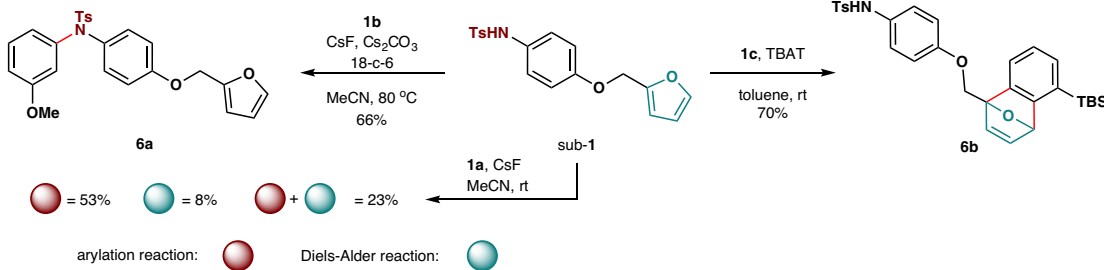

both the high electrophilicity[30] and fleeting property of an aryne species, unfortunately, to govern aryne reactions chemoselectively on polyfunctionalized substrates has been traditionally seen as an almost impossible task. Although aryne intermediate can occasionally distinguish a reacting site from the others on a complex molecule, this approach heavily relies on the inherent reactivity difference among those arynophiles and frequently encounters with unavoidable side products from competing reactions whenever this difference is not decisive (Fig. 1B). For instance, a distinct work was reported by Hoye and Ross, in which hexadehydro-Diels–Alder (HDDA) reaction-derived arynes could undergo site-selective transformations on several structurally complex natural products bearing multiple potential reaction

sites[31]. Recently, McCormick et al. developed an arynophilicity parameter to describe the relative reactivity of various arynophiles with respect to 3-chlorobenzyne[32]. Besides, there are some other competitive aryne reaction examples[33,34].

It has been commented by Hergenrother that "Can robust predictive models for site- and chemoselectivity of this and other (aryne) methodologies be developed? and eventually enable the ultimate goal of orthogonal functionalization of any site of a complex natural product?"[35]. Unfortunately, there is no doubt that it would be very challenging to govern chemoselectivity at will by using highly reactive and fleeting aryne species. This fundamental topic becomes even emerging in view of the rapid recent discovery of benzyne transformations as well as the potential application of these useful synthetic tools. Here, chemoselective control between two major classes of aryne reactions, namely nucleophilic-type reactions and pericyclic reactions, was developed by using 3-substituted arynes, which employs both electronic effect and steric repulsion to differentiate the reaction rates of a collection of arynophiles. Arynes containing 3-electron-withdrawing groups (3-EWGs) were found to preferentially react with nucleophiles on its C1-position in the presence of non-polar pericyclic reaction partners, the position of which is more electrophilic and less sterically encumbered. Alternatively, an inverse chemoselectivity in favor of pericyclic reactions was unraveled when 3-(*tert*-butyldimethylsilyl)benzyne (3-(TBS)benzyne) was utilized, whereas nucleophilic addition reactions were prohibited due to the counteracted electronic and steric effect on both the C1- and C2-positions of 3-silylbenzyne. Particularly notable is the power of this protocol to inversely differentiate two types of arynophiles that possess similar reactivity toward simple benzyne.

## Results

Along with our study on benzyne chemistry[12], we were curious about the possibility to discriminate arynophiles of different types, i.e., polar and non-polar ones, by unraveling some seemingly trivial reactivity discrepancy among those highly reactive aryne species. In order to see whether different aryne species behave consistently or not in the presence of various types of arynophiles, we decided to assess the competing reaction of imidazole (**2a**) and furan (**3a**) with them. As shown in Fig. 1C, it was found that the trapping reaction of simple benzyne precursor **1a** with excess amounts of imidazole (**2a**) (3.0 equiv) and furan (**3a**) (3.0 equiv) afforded the N-arylation product **4a** in 36% yield and the [4 + 2]-cycloadduct **5a** in 47% yield with a ratio of 1:1.3 (entry 1). This ratio could change to 2.1:1 when 18-c-6 was used, suggesting that the reaction conditions could slightly alter the product ratio (entry 2). When 3-methoxybenzyne precursor **1b** was employed, **4b** was obtained in 74% yield along with only 4% of cycloadduct **5b** with a ratio of 19:1 (entry 3). By contrast, other aryne precursors containing 3-F or 3-Br substituent could not provide satisfied product ratio (for details, please see Supplementary Table 1). Next, we turned our attention to arynes bearing 3-electron-donating groups (3-EDGs). Among an array of 3-silylbenzyne precursors (Supplementary Table 1), 3-(TBS)benzyne precursor **1c** was found to be the most favorable one, featuring cycloadduct **5c** as the major product in 72% yield and **4c** in 7% yield with a ratio of 10:1 (entry 4). Note that precursors of both 3-chlorobenzyne **1d** (entry 5), which was used by McCormick and Stuart in their arynophilicity study[32], and 3-(trimethylsilyl)benzyne **1e** (entry 6) could not gave satisfied product ratio in comparison to the corresponding **1b** and **1c**, respectively. In addition, 3-*tert*-butylbenzyne precursor **1f** was tested and its reaction furnished both **4f** and **5f** in a low product ratio (1.3:1), indicating that steric repulsion itself cannot serve as the decisive factor to differentiate these two types of aryne reactions (entry 7).

Beyond imidazole and furan, the same type of switchable chemoselectivity was observed between an array of nucleophiles, i.e., N-tosylated aniline (**2b**), N-benzyl aniline (**2c**), phenol **2d**, benzyl alcohol (**2e**), and benzoic acid (**2 f**), and pericyclic reaction partners, i.e.,

cyclopentadiene (**3b**), N-tosyl pyrrole (**3c**), and 1-phenylcyclohexene (**3d**) (Fig. 1D and please also see Supplementary Table 2). These examples illustrate that the designated chemoselectivity could be accomplished between appropriate nucleophile and pericyclic reaction partner. The key tuning factor is by simply switching the 3-substituent, either a methoxy group or a TBS group, on the parent benzyne ring. Notably, the closer the reactivity toward simple benzyne between two types of arynophiles, the more effective tuning ability on both reaction directions with respect to 3-methoxybenzyne and 3-(TBS)benzyne could be reached (Fig. 1D).

Our next goal is to apply this chemoselective control protocol to molecules containing both nucleophilic and pericyclic components, which possesses more practical potential than those intermolecular cases in Fig. 1D. As shown in Fig. 1E, substrate-**1** (sub-**1**) was chosen. Comparing with intermolecular reaction conditions, the conditions for compound sub-**1** are apparently different. For instance, stoichiometric or excess amount of aryne precursor is necessary, which in turn markedly raises the risk of aryne to interact with unwanted reaction sites. When 1.5 equiv of 3-methoxybenzyne precursor **1b** was used to react with sub-**1**, compound **6a** could be obtained solely in 66% yield. Inversely, the reaction between **1c** and sub-**1** by using tetra-butylammonium difluorotriphenylsilicate (TBAT) in toluene afforded the cycloadduct **6b** in 70% yield with almost no detectable amount of the N-arylation product. In contrast, the reaction of sub-**1** with simple benzyne precursor **1a** gave rise to a mixture of N-phenylation product (58%), cycloadduct (8%), and a compound resulted from both N-phenylation and [4 + 2]-cycloaddition (23%) (Fig. 1E). Notably, reverse chemoselectivity could be realized on a molecule by employing properly chosen 3-substitued arynes.

Alike compound sub-**1**, both the N-arylation product **7a** (Fig. 2A) and the cycloadduct **7b** (Fig. 2B) were obtained from compound sub-**2** (Fig. 2C) when treating with either **1b** or **1c**, respectively. Moreover, N-alkyl anilines as more prevailing functional groups appeared to be appropriate nucleophilic partners with furan. Accordingly, the N-arylation products **8a**, **9a**, and **10a** were obtained from substrates sub-**3**, sub-**4**, and sub-**5**, respectively, in good to high yields, giving an alternative approach for arene amination (Fig. 2A)[36–39]. Alternatively, when the same substrates were treated with aryne precursor **1c**, the corresponding cycloadducts **8b**, **9b**, and **10b** were isolated in high overall yields as mixtures of diastereoisomers (Fig. 2B). By enhancing the steric congestion around the N-nucleophile of anilines, improved product ratio could be achieved, suggesting that steric effect plays an essential role in tuning the product ratio as well. Next, Fuberidazole, a compound used in fungicides, containing a furan and a benzimidazole subunit was investigated. Its reaction with **1b** produced only the N-arylation product **11a** in high yield; while chemoselective [4 + 2]-cycloaddition reaction was achieved when **1c** was utilized, furnishing **11b** and **11b'** in 86% overall yield as a mixture of 1:1 regioisomers (Fig. 2B). Pyrrole was then considered in our study. Both tosylamide and phenol were found to serve as appropriate reaction partners of pyrrole group. To this end, **12a** and **13a** were obtained from sub-**6** and sub-**7** in good yields after arylation with **1b** in the presence of carbonate salts (Fig. 2A); while the treatment of **1c** delivered the corresponding [4 + 2]-cycloadducts **12b** and **13b** exclusively (Fig. 2B). Next, the reaction of compound sub-**8** bearing a cyclopentadiene and two phenol moieties with excess amount of **1b** produced the double O-arylation product **14a** in 60% yield. Alternatively, when **1c** was employed, the [4 + 2]-cycloadduct **14b** could be achieved solely in 70% yield. In addition, substrate sub-**9** bearing an alcohol and a cyclohexenyl group was tested, in which both reaction sites were found to possess comparable reaction rate toward benzyne (Fig. 1D). The corresponding O-arylation product **15a** and the ene reaction product **15b** were obtained by treating sub-**9** with **1b** and **1c**, respectively, in excellent chemoselectivity. In order to demonstrate the merit of both **1b** and **1c** as distinctive aryne precursors, all the substrates in Fig. 2D

were examined with simple benzyne precursor **1a**. In sharp contrast to Fig. 2A, B, mixtures of products attributed from single N/O-arylation, single pericyclic reaction, and/or arylation-pericyclic dual reactions were usually isolated (Fig. 2C).

Furthermore, *N*-deacetyl *N*-tosyl Colchicine sub-**10** was prepared, which contains a tosylamide and a cycloheptatrienone subunit (Fig. 3A). The reaction of sub-**10** with **1b** produced the N-arylation product **16a** only in 66% yield, while the employment of aryne precursor **1c** furnished

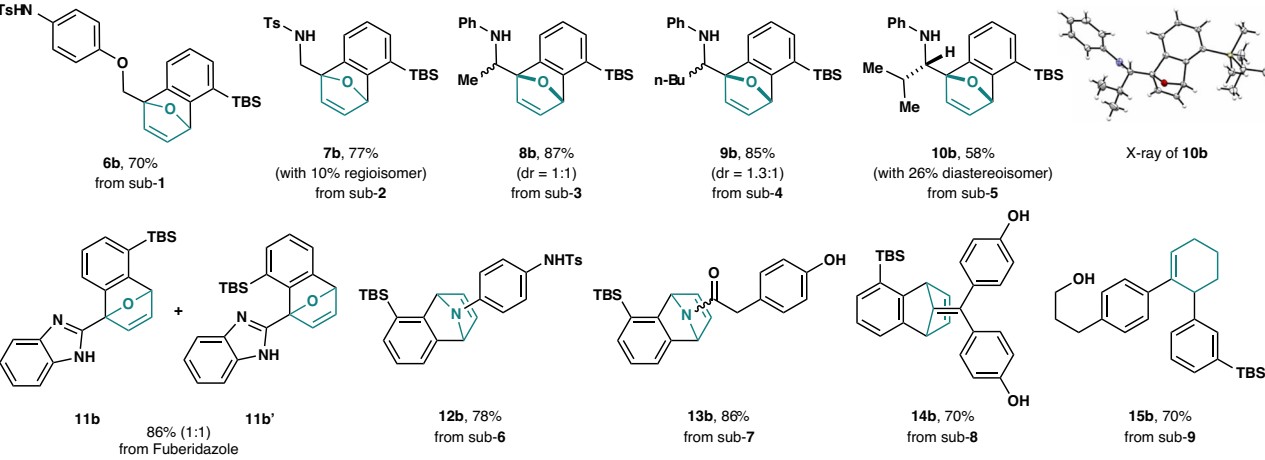

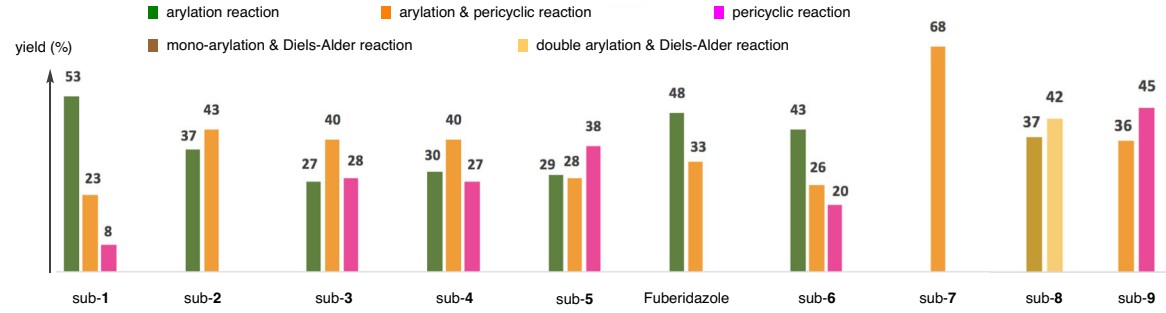

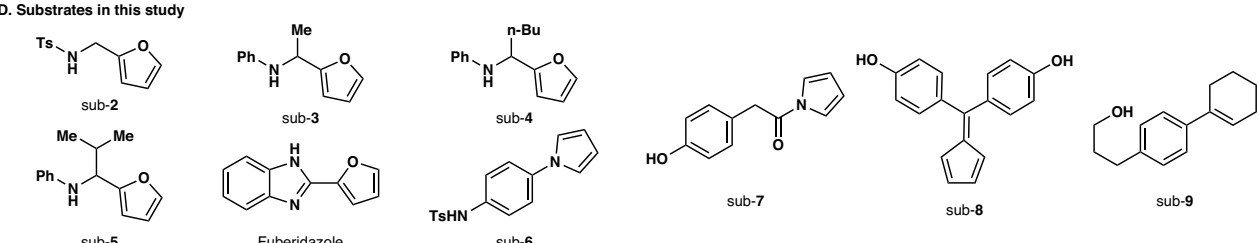

**Fig. 2 | Study on intramolecular chemoselective aryne reactions. A** Selective nucleophilic addition reactions with 3-methoxybenzyne precursor **1b**. **B** Selective pericyclic reactions with 3-(TBS)benzyne precursor **1c**. **C** Reaction products with simple benzyne precursor **1a**. **D** Substrates in this study.

**Fig. 3 | Further investigation on intramolecular systems. A** The reactions of *N*-deacetyl *N*-tosyl Colchicine sub-**10** with **1b** and **1c**. **B** Selective N-arylation with other aryne precursors. **C** Further conversion of the TBS group on the [4 + 2]-cycloadducts.

the [4 + 2]-cycloadduct **16b** in 78% yield with distinct regio and facial selective control. Beside 3-methoxybenzyne as preferential arylating reagent, several other aryne species were found to serve as the same role to react with compound sub-**5**, giving rise to **17a**–**17d** in good yields (Fig. 3B). Following Han's protocol[40], the TBS group on **10b** could be selectively removed in the presence of *t*-BuOK in DMSO, furnishing **18** in excellent yield (Fig. 3C). Alternatively, both cycloadducts **11b** and **11b'** could be converted to naphthalene **19** by treating them with chlorotrimethylsilane (TMSCl) and sodium iodide (NaI) in acetonitrile[41], the transformation of which experiences a deoxygenative aromatization and a desilylation process. Both cases in Fig. 3C demonstrate that the TBS group on the [4 + 2]-cycloadduct could be readily removed, making 3-(TBS)benzyne an equivalent of simple benzyne with the ability to differentiate cycloaddition reactions from nucleophilic addition reactions.

In order to expand the scope of our protocol from arylation reactions to other types of transformations, compound sub-**11** containing a trifluoroacetamide and a furanyl group was prepared, the trifluoroacetamide site of which is known to undergo a C–N bond insertion reaction with benzyne[42]. While the reaction of sub-**11** with **1b** could feature the C–N bond insertion product **20a** in 67% yield, the employment of **1c** produced the cycloadduct **20b** in 81% yield accompanied with 10% of its regioisomer (Fig. 4A). We then prepared compound sub-**12** to see whether a Truce–Smiles rearrangement could serve as an appropriate partner of furan cycloaddition reaction[43]. Treating sub-**12** with **1b** in refluxing THF gave rise to the desired Truce–Smiles rearrangement product **21a** in 73% yield, while the furanyl group kept inert even under elevated temperature (Fig. 4A). Alternatively, the reaction between sub-**12** and **1c** in DCM afforded the cycloadduct **21b** in 57% yield along with 14% of its regioisomer. Another example is compound sub-**13**, which contains a Diels–Alder partner pyrrole and a methyl salicylate component that could undergo nucleophilic annulation reaction (Fig. 4A). The reaction of sub-**13** with **1b** in refluxing THF delivered the nucleophilic annulation product **22a** in 72% yield, while cycloadduct **22b** could be obtained in 80% yield as a 1.7:1 mixture of conformational isomers in the reaction with **1c**. The examples in Fig. 4A demonstrate that not only typical arylation reactions, but also a range of nucleophilic addition-triggered transformations can be applicable to our chemoselective control protocol.

Moreover, we wanted to see whether this strategy can be applied in more sophisticated molecular systems, such as those bearing more than two reaction sites. Furosemide, a loop diuretic medication used to treat fluid build-up due to heart failure, liver scarring, or kidney disease, was chosen, which contains four potential reaction sites, a pericyclic partner furan and three nucleophiles. Due to the low solubility of the zwitterionic structure of Furosemide, its ester derivative sub-**14** was prepared for our study (Fig. 4B). It was found that the reaction of Furosemide methyl ester sub-**14** with **1b** delivered the double nucleophilic addition product **23a** exclusively on the sulfonamide. The absence of nucleophilic addition on the secondary amine of sub-**14** should be attributed to the steric effect of its *ortho*-ester group. Alternatively, selective Diels–Alder reaction on the furanyl group of sub-**14** occurred with **1c**, featuring compound **23b** in 78% yield along with 12% of its regioisomer. Next, compound sub-**15** bearing two types of N-nucleophiles and a furfuryl group was prepared, where a different reactivity sequence from sub-**14** was disclosed (Fig. 4B). In this case, the aniline nitrogen on sub-**15** was found to be more reactive than both benzimidazole nitrogen and furan. To this end, simple benzyne precursor **1a** was employed to capture it, producing compound **24** in 78% yield. At this stage, the diaryl tertiary amine subunit on **24** turned out to be less nucleophilic than benzimidazole. Consequently, both selective N-arylation on the benzimidazole with **1b** and [4 + 2]-cycloaddition on the furan with **1c** could then be realized on compound **24**, affording the corresponding products **25a** and **25b**, respectively. Compound sub-**16** containing a cyclopentadiene, a primary alcohol, and a tertiary alcohol was then prepared. The reaction of sub-**16** with **1b** featured the O-arylation product **26a** in 74% yield, while its reaction with **1c** produced the cycloadduct **26b** in 78% yield.

To demonstrate the synthetic potential of this chemoselective strategy, an asymmetric preparation of dihydrexidine, an agonist at the dopamine D1 and D5 receptors with antiparkinson effects, was developed (Fig. 5)[44]. Starting from Sesamol, a TBS group was incorporated onto its 2-position in a one-pot process, featuring compound **27** in 85% yield. Next, compound **28** was obtained in 83% yield upon bromination, which was converted to aryne precursor **29** following the standard procedure. Meanwhile, compound **30** was readily prepared via Suzuki-Miyaura coupling reaction and reduction, which could then

**A. Study on other nucleophilic addition-triggered reactions**

**B. Substrates with more than two reaction sites**

**Fig. 4 | Study on more complex systems. A** Nucleophilic addition-triggered transformations. The investigation on C–N bond insertion, Truce–Smiles rearrangement, and nucleophilic annulation reaction suggests that this chemoselective protocol can be utilized in a broad range of aryne reaction modes. **B** Study on substrates bearing more than two reaction sites. When an arynophile is much more reactive toward benzyne, it can be trapped by simple benzyne so that subsequent chemoselective control between other arynophiles with matched reactivity could occur. By contrast, a less reactive reaction site will keep inert with respect to any aryne species.

**Fig. 5 | Synthetic application.** Asymmetric synthesis of dihydrexidine hydrobromide salt.

be transformed to compound **31** in two steps. Chemoselective Diels–Alder reaction between **29** and **31** furnished cycloadducts **32a** and **32b** as a mixture of 1:1.2 regioisomers in 90% overall yield, whereas almost no nucleophilic addition reaction was observed. The TBS group on **32a** and **32b** could then be easily removed upon treating with *t*-BuOK in DMSO, affording compound **33** in 90% yield. Notably, this two-step operation could facilely convert compound **31** to **33** in an overall excellent yield. Next, rhodium-catalyzed cycloisomerization on oxabicycle **33** was examined. Inspired by Lautens' previous study[45], kinetic resolution of racemic oxabicycle **33** in the presence of the (R,S)-PPF-P'Bu2 Josiphos ligand gave rise to compound **34** in 30% yield and 98% ee (Fig. 5). The absolute configuration of compound **34** was assigned by its x-ray single crystal analysis. With enantiopure **34** in hand, hydrogenation, followed by dehydroxylation reaction by using triethylsilane and boron trifluoride etherate, generated compound **35** in 44% yield, accompanied with 45% of its diastereoisomer. Finally, deprotection of both the acetal and the tosylate groups on **35** delivered dihydrexidine hydrobromide salt in 86% yield.

To elucidate the origin of the above chemoselectivity, we decided to compare the reaction rates of several 3-substituted arynes in the presence of both *n*-butylamine and *t*-butylamine. In the study on 3-silylbenzyne carried out by both the groups of Akai[46] and Garg[47], they disclosed that the size of a nucleophile could significantly influence the preferred site of attack, where a small nucleophile prefers the electronically favored *ortho*-position of the silyl group and a bulky nucleophile inclines to attack the electronically disfavored *meta*-position. When **1e** was treated with both *n*-butylamine (3.0 equiv) and *t*-butylamine (3.0 equiv), **36a**, **36b**, and **36c** were isolated in 68%, 13%,

and 7% yields, respectively (Fig. 6A). Moreover, the employment of **1c** produced **37a** in 44% yield, **37b** in 40% yield, and **37c** in 3% yield, which is consistent with the fact that TBS group is more sterically congested than TMS group. In comparison, a competing reaction with 3-methoxybenzyne precursor **1b** was carried out, furnishing **38a** in 60% yield and **38b** in 32% yield (Fig. 6A). These experiments demonstrate that although *t*-butylamine is slightly less efficient than that of *n*-butylamine in the reaction with **1b**, its reactions with **1e** and **1c** are suppressed by the presence of *n*-butylamine. In this context, a σ-electron-donating silyl group (in our study the TBS group) on the C3-position of a benzyne not only perturbs benzyne distortion to favor *ortho*-attack with respect to small nucleophiles, but also slows down the rate of nucleophilic addition by bulky nucleophiles when *meta*-selectivity becomes demanding due to steric repulsion on its *ortho*-position (Fig. 6B). Whenever an arynophile is not heavily influenced by the aryne distortion imposed by the 3-silyl group, chemoselective control should be realized. By contrast, 3-methoxybenzyne would undoubtedly favor nucleophilic-type reactions on its C1-position, because this site is both electronically favored and sterically less congested. Of course, the success of this protocol also depends on the magnitude of nucleophilicity of an arynophile. In this scenario, either a strong nucleophile would by no means to attack 3-silylbenzyne over other arynophiles or a weak nucleophile should keep inert throughout the reaction (Fig. 4B).

Next, Density functional theory (DFT) calculations were carried out at the M062x/6-31 + G(d) level of theory. Our calculation indicates that simple benzyne could not differentiate the reaction with either imidazole (**2a**) or furan (**3a**) due to close activation energies between

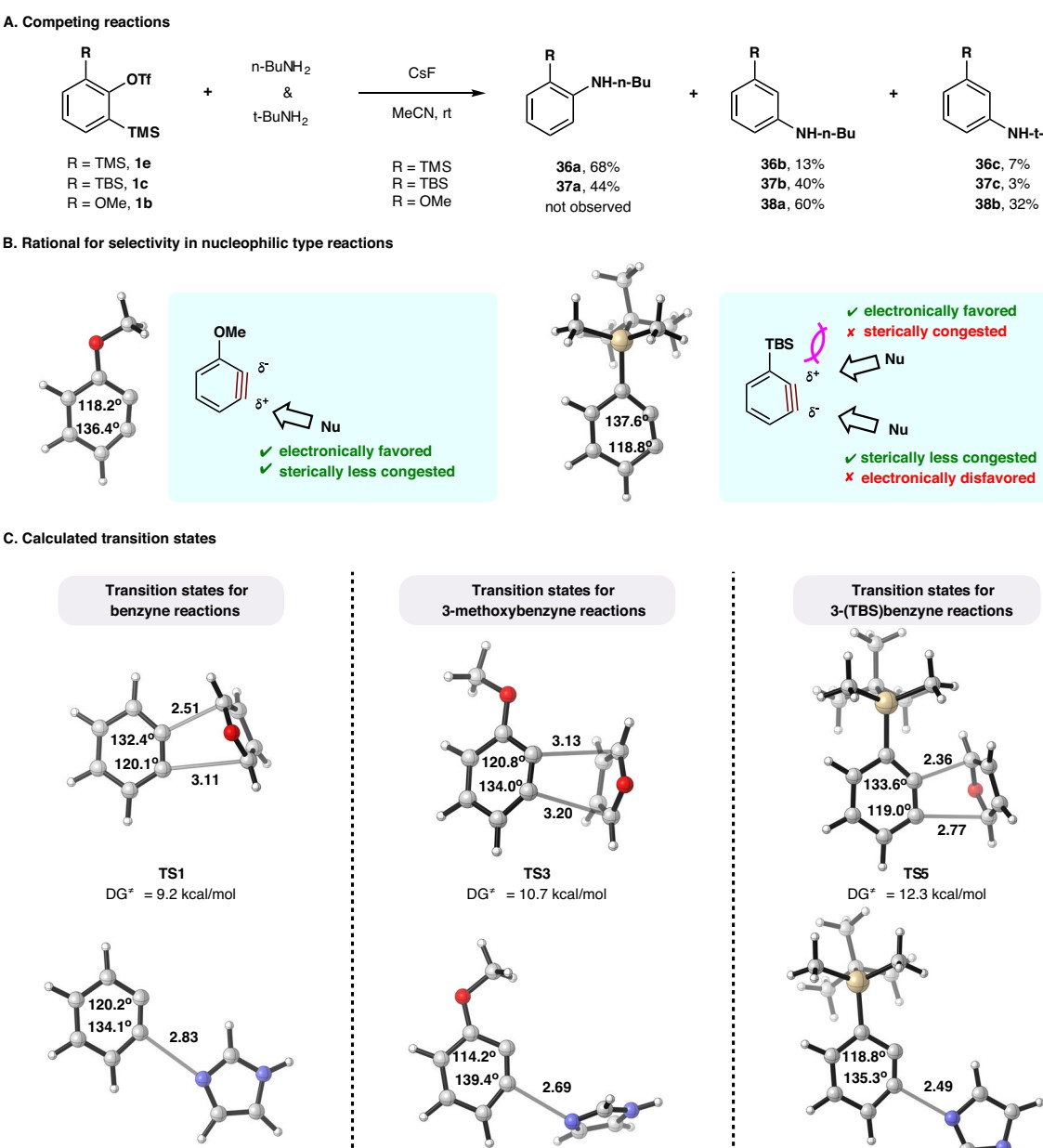

**Fig. 6 | Mechanistic investigation. A** Competing reactions of aryne precursors with *n*-butylamine and *t*-butylamine. **B** Rational for selectivity in nucleophilic type reactions on either 3-methoxybenzyne or 3-(TBS)benzyne. **C** Transition states for the reactions of benzyne, 3-methoxybenzyne, and 3-(TBS)benzyne with either furan and imidazole. The free energy values are calculated at the M06-2X/6-31 + G(d) level of theory in acetonitrile with SMD model using the Gaussian 09 series of programs. Hydrogen, white; Carbon, gray; oxygen, red; silicon, yellow; nitrogen, purple.

transition states **TS1** and **TS2** (9.2 kcal/mol vs. 9.4 kcal/mol) (Fig. 6C). Indeed, the reactions with 3-methoxybenzyne prefers N-arylation reaction due to lower activation energy for **TS4** than that for **TS3**; whereas the energy difference between transition states of [4 + 2]-cycloaddition **TS5** and N-arylation **TS6** with respect to 3-(TBS)benzyne is 1.4 kcal/mol in favor of the cycloaddition reaction (Fig. 6C). Overall, three factors ensure the success of this chemoselective aryne reaction strategy: a. the inherent reactivity difference between a nucleophile and a pericyclic reaction partner determines the probability for chemoselective control; b. proper 3-substituent on benzyne could modulate the preferred reaction mode; c. additional factors, such as steric effect and reaction conditions, further enhance the product ratio.

We have been able to demonstrate that chemoselective control between two major types of benzyne transformations, namely

nucleophilic addition-triggered reactions and pericyclic reactions, could be accomplished by simply varying the 3-substituent on aryne intermediate. In the presence of appropriate nucleophile and pericyclic reaction partner, arynes containing 3-EWG groups incline to take part in nucleophilic addition reactions. Alternatively, 3-(*tert*-butyldimethylsilyl)benzyne slows down nucleophilic addition reaction on both its triple bond carbons due to a mismatch between electronic effect and steric repulsion and, hence, results in preferential pericyclic reaction. This protocol is amenable to a broad range of both nucleophilic addition reactions, i.e., transition metal-free arylation reactions, insertion reaction, Truce–Smiles rearrangement, and nucleophilic annulation reaction, and pericyclic reactions, i.e., Diels–Alder reaction and ene reaction, which unravels a great potential of aryne chemistry in manageable chemoselective functionalization of complex

molecules. It is expected that more general and effective chemoselective strategies can be unraveled in the future.

## Methods

### General procedure for the reaction of benzyne precursor 1a

A mixture of benzyne precursor **1a** (89.5 mg, 0.3 mmol, 1.5 equiv), substrate (0.2 mmol, 1.0 equiv), and CsF (91.1 mg, 0.6 mmol, 3.0 equiv) in MeCN (2.0 ml) under inert atmosphere was stirred at room temperature overnight. The resulting mixture was filtered through a short pad of silica gel (EtOAc eluent, 20 ml). All the volatiles were removed on a rotary evaporator. The crude material was purified by flash column chromatography to afford the corresponding products.

### General procedure for the reaction of aryne precursor 1b

A mixture of aryne precursor **1b** (98.5 mg, 0.3 mmol, 1.5 equiv), substrate (0.2 mmol, 1.0 equiv), CsF (91.1 mg, 0.6 mmol, 3.0 equiv), and 18-c-6 (105.7 mg, 0.4 mmol, 2.0 equiv) in anhydrous MeCN or THF (2.0 ml) under inert atmosphere was stirred overnight. The resulting mixture was filtered through a short pad of silica gel (EtOAc eluent, 20 ml). All the volatiles were removed on a rotary evaporator. The crude material was purified by flash column chromatography to afford the corresponding products.

### General procedure for the reaction of aryne precursor 1c

A mixture of aryne precursor **1c** (123.8 mg, 0.3 mmol, 1.5 equiv), substrate (0.2 mmol, 1.0 equiv), KF (46.5 mg, 0.8 mmol, 4.0 equiv), and 18-c-6 (105.7 mg, 0.4 mmol, 2.0 equiv) in anhydrous THF, MeCN, or DCM (2.0 ml) under inert atmosphere was stirred at room temperature overnight. The resulting mixture was filtered through a short pad of silica gel (EtOAc eluent, 20 ml). All the volatiles were removed on a rotary evaporator. The crude material was purified by flash column chromatography to afford the corresponding products.

## Data availability

The data generated in this study are provided in the Supplementary Information/Source Data file. Details about materials and methods, experimental procedures, characterization data, and NMR spectra are available in the Supplementary Information. Crystallographic data for the structures reported in this article have been deposited at the Cambridge Crystallographic Data Centre under deposition numbers CCDC 2234236 (**10b**) and 2288207 (**34**). Copies of the data can be obtained free of charge via https://www.ccdc.cam.ac.uk/structures/. All data are available from the corresponding author upon request. Source data are provided with this paper.

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

## Acknowledgements

The authors gratefully acknowledge research support of this work by NSFC 21971028 (Y.L.), 22325104 (Y.L.), 21772017 (Y.L.), 21901025 (J.S.), 22103008 (C.S.).

## Author contributions

H.T., S.Y., X.Y., L.C. and J.S. performed experiments and analyzed experimental data. C.S. did the DFT calculations. Y.L. directed the project and prepared the manuscript.

## Competing interests

The authors declare no competing interests.
