## [Peer Review File · Nature Communications]

Switchable Chemoselective Aryne Reactions between Nucleophiles and Pericyclic Reaction Partners using either 3-Methoxybenzyne or 3-SilylbenzyneREVIEWER COMMENTS

Reviewer #1 (Remarks to the Author):

In this paper, Li and coworkers described a comprehensive study on the chemoselectivity of the reaction of arynes with substrates containing two reaction moieties. This study demonstrated the arynophilicity parameter (ref. 33) depends on the structure of arynes and provided guidelines for selecting an appropriate aryne precursor in reactions with complex molecules. The wide substrate scope demonstrates the versatility of this approach and Supporting Information is carefully prepared. This reviewer considers this paper contains a novel concept on aryne chemistry achieving chemoselective reactions based on the orthogonal reactivities of 3-substituted arynes and arynophiles/nucleophiles. In view of these, this reviewer recommends this manuscript to be published in Nature Communications after addressing several following points.

1. Some readers might misinterpret this study is focusing on the mechanism of chemoselectivity of each aryne precursor, which is already predictable from the previous reports (Ref. 28–30, 33). Because both the aryne and the arynophile/nucleophile are substrates, changing "substrate-controlled" in the title is recommended. As the key of this study is the discovery of orthogonal reactivities between 3-methoxyaryne and 3-silylarynes toward arynophiles for pericyclic reactions and nucleophiles for arylations, more detailed explanation should be included in the abstract and the title should also focus on this finding. Switchable chemoselectivity is achieved by changing the substituent (methoxy or silyl group) at the 3-position of arynes.
2. More detailed explanation on how the authors chose the reaction conditions, particularly what kind of situation is better for employing Cs₂CO₃ for the reactions using 3-methoxybenzene precursor 1b, should be provided.
3. References should be selected more carefully. For example, this reviewer suggests citing 27 reports to highlight the utility of arynes may be excessive. Instead, the authors should mention other important papers that conduct competitive experiments using two arynophiles (eg. Chem. Pharm. Bull. 2022, 70, 566). It is also recommended to cite some papers performing reactions of arynes bearing functional groups that can react with arynes (eg. Chem. Lett. 2022, 51, 94).
4. Authors should be added for refs 16, 17, 20, 30, and 41.
5. Supporting Information:
 - a) Page S24: "only product" should be "sole product", and "1:<20" should be "1:>20".
 - b) Describe how the stereochemistry of the diastereomers was determined, such as S45/S46, S49/S50, S53/S54, S55/18, and S81/35.

Reviewer #2 (Remarks to the Author):

Li and coworkers present a chemoselective strategy between nucleophiles and dienes by using different 3-substituted arynes. In the presence of 3-silyl substituted arynes, cycloaddition reactions occur preferentially over nucleophilic addition reactions, and the employment of 3-methoxybenzyne could switch the chemoselectivity toward nucleophilic addition reactions. The authors examined a variety of functional groups, such as imidazole, N-tosylated/N-alkyl aniline, phenol, alcohol, furan, cyclopentadiene, pyrrole, cycloheptatrienone, and cyclohexene, and found that this strategy could be applied in other types of transformations, such as C–N bond insertion, Truce–Smiles rearrangement, and nucleophilic annulation. Therefore, the substrate scope of this method is broad. Then, the

authors give a synthesis of dihydrexidine as a demonstration for the application of their chemoselective strategy. The key steps are a selective 4+2 cycloaddition reaction and the removal of the TBS group, which should inspire people for the potential application of this strategy. At last, they use competing reactions and DFT calculations to understand the origin of the chemoselectivity, which is reasonable.

Although it is known that an electron-withdrawing group on the 3-position of benzyne could control nucleophile to attack its meta position in regioselective manner, interestingly, this study is quite different by controlling the chemoselectivity between nucleophile and dienes. Moreover, it is unexpected that 3-silyl substituted arynes could be able to pick up dienes from various nucleophiles. This discovery has not been previously revealed. Although there are numerous successful examples on new transformations of benzyne, it is the first time that Li could control chemoselectivity in benzyne reactions. Overall, this reviewer thinks that the study by Li and coworkers is an inspiring and deep investigation. Both the manuscript and the supporting information are well prepared. This reviewer believes that this work is an interesting discovery in benzyne chemistry and meets the requirement for novelty by Nat. Commun. and should be accepted after minor revision.

Questions:

- 1) Have the authors tried to examine other types of cycloaddition reactions, such as 3+2 and 2+2 cycloaddition reactions?
- 2) Could 3-silyl substituted arynes be used to select a nucleophile among several different nucleophiles? This would be interesting.
- 3) Could the authors explain why 3-substituted arynes other than those reported in Fig. 3b are not applicable in this study?
- 4) As an ultimate goal, it would be great to see chemoselectivity by using simple benzyne. Would it be possible to use unsubstituted benzyne to realize chemoselectivity between nucleophile and dienes?
- 5) Figure 2A, 15a was derived from a primary alcohol, how about the reaction results when secondary alcohols and tertiary alcohols were utilized?
- 6) This work is involved in arene amination, some related work should be cited in a suitable position (J. Am. Chem. Soc. 2017, 139, 15656; J. Am. Chem. Soc. 2018, 140, 4503; Sci. Chin. Chem. 2024, 67, 374; ACS Catal. 2020, 10, 8402; Angew. Chem. Int. Ed. 2019, 58, 10245).

Reviewer #3 (Remarks to the Author):

Recommendation: Minor Revisions required

Comments:

In this work, Li and coworkers reported a substrate-controlled switchable chemoselective aryne reaction between nucleophiles and pericyclic arynophiles. Authors developed manipulate chemoselective control methods between two major types of aryne transformations. Nucleophilic addition-triggered reactions and non-polar pericyclic reactions could be differentiated by employing different substituted arynes. This substrate controlled chemoselective protocol was found to be applicable between various nucleophiles, the applications showed that the method had a good perspective in the field of synthetic application. Control experiments were designed and carried out carefully. The manuscript was well organized, and should be interesting to the general readership in the field of synthetic chemistry. Thus, I agree to the publication of this manuscript in Nature

Communications after addressing the following minor points.

1. As for the subsequent research results corresponding to the conditional screening given in Fig. 1c, most insertion reactions require the participation of 18-c-6. But when 3-(TBS) benzene precursors occur [4+2] cycloaddition reactions, 18-c-6 is hardly required. However, the compounds 20b, 21b, 22b and sub-15 all required the participation of 18-c-6. Did it conflict with each other? What role did 18-c-6 play in the reaction?

2. When the reaction of 3-substituted aromatic hydrocarbons to nucleophilic and pericyclic aromatic hydrocarbons were controlled chemically, did the choice of reaction solvent affect the selectivity of the reaction? If so, can you summarize the general pattern? How did different types of bases affect the reaction?

Reviewer #1 (Remarks to the Author):

In this paper, Li and coworkers described a comprehensive study on the chemoselectivity of the reaction of arynes with substrates containing two reaction moieties. This study demonstrated the arynophilicity parameter (ref. 33) depends on the structure of arynes and provided guidelines for selecting an appropriate aryne precursor in reactions with complex molecules. The wide substrate scope demonstrates the versatility of this approach and Supporting Information is carefully prepared. This reviewer considers this paper contains a novel concept on aryne chemistry achieving chemoselective reactions based on the orthogonal reactivities of 3-substituted arynes and arynophiles/nucleophiles. In view of these, this reviewer recommends this manuscript to be published in Nature Communications after addressing several following points.

Reply: we thank this referee for these comments.

1. Some readers might misinterpret this study is focusing on the mechanism of chemoselectivity of each aryne precursor, which is already predictable from the previous reports (Ref. 28–30, 33). Because both the aryne and the arynophile/nucleophile are substrates, changing "substrate-controlled" in the title is recommended. As the key of this study is the discovery of orthogonal reactivities between 3-methoxyaryne and 3-silylarynes toward arynophiles for pericyclic reactions and nucleophiles for arylations, more detailed explanation should be included in the abstract and the title should also focus on this finding. Switchable chemoselectivity is achieved by changing the substituent (methoxy or silyl group) at the 3-position of arynes.

Reply: thanks for this suggestion. Accordingly, we changed title to "Switchable Chemoselective Aryne Reactions between Nucleophiles and Pericyclic Arynophiles using either 3-Methoxybenzyne or 3-Silylbenzyne". Meanwhile, we made changes to the abstract as well.

2. More detailed explanation on how the authors chose the reaction conditions, particularly what kind of situation is better for employing Cs₂CO₃ for the reactions using 3-methoxybenzene precursor 1b, should be provided.

*Reply: all benzyne reactions are not single-step transformations. They all include a benzyne generation step. Due to the high reactivity and short lifetime of a benzyne intermediate, slow releasing of benzyne from its precursor is necessary. Therefore, different reaction conditions including various solvents, fluoride salts, and/or 18-c-6 are normally used in order to tune the generation rate of benzyne so that it can match well with the consequent reactions. In our study, we examined different types of reaction modes with quite different substrates. For each substrate, we had to screen several reaction conditions so that it could reach the optimal selectivity and yield. Therefore, there is no definite reaction conditions that could cover different types of substrates in our study. To response this question by this reviewer, we added general procedures for the reactions with those aryne precursors to the end of our manuscript, where they could be applied to most of the substrates. As for the role of bases in the reactions of **sub-1**, **sub-6**, and **sub-7** (Fig 2) with 3-methoxybenzyne, the bases could deprotonate those nucleophiles so as to increase their nucleophilicity.*

3. References should be selected more carefully. For example, this reviewer suggests citing 27 reports to highlight the utility of arynes may be excessive. Instead, the authors should mention other important papers that conduct competitive experiments using two arynophiles (eg. Chem. Pharm. Bull. 2022, 70, 566). It is also recommended to cite some papers performing reactions of arynes bearing functional groups that can react with arynes (eg. Chem. Lett. 2022, 51, 94).

Reply: we removed ref 27 and added the recommended two references (ref 33 and 34). Thanks for these suggestions.

4. Authors should be added for refs 16, 17, 20, 30, and 41.

Reply: thanks for reminding this. In the Nature formatting guide, it indicates that “All authors should be included in reference lists unless there are more than five, in which case only the first author should be given, followed by ‘et al.’.” For those references, there are more than five authors.

5. Supporting Information:

a) Page S24: “only product” should be “sole product”, and “1:<20” should be “1:>20”.

Reply: thanks. we made the changes accordingly.

b) Describe how the stereochemistry of the diastereomers was determined, such as S45/S46, S49/S50, S53/S54, S55/18, and S81/35.

Reply: the *J* value for **S81** is 5.2 Hz at 3.82 ppm, and the *J* value for compound **35** is 11.2 Hz at 4.05 ppm, suggesting that compound **35** is *trans*-isomer. Besides, compound **35** was converted to dihydroxidine hydrobromide, and the NMR of the product is identical to the known compound.

We obtained the X-ray crystal structure of compound **10b** (Fig 2B), and the TBS group on **10b** was readily removed to produce **18** (Fig 3C). We also did phenylation reaction on compound **18** with simple benzyne precursor and obtained compound **S54**.

On the other hand, in the reaction between **sub-5** and **1c**, we obtained **10b'**, a stereoisomer of **10b**, in 26% yield (see Fig 12B). The desilylation of **10b'** obtained **S55** and the consequent phenylation gave rise to **S53**.

As for *S45/S46* and *S49/S50*, since *sub-3* and *sub-4* are the structural analogues of *sub-5*, their cycloaddition reactions with benzyne should give the same stereoselective control with those for *sub-5*. However, the facial selectivity in these reactions is not distinct. As shown in the following figure, the ratio of *S46/S45* is 2.3:1 and that of *S50/S49* is 3:1, both have close value with the ratio of *S54/S53*. Since methyl group is smaller than iso-propyl group, it is reasonable that *S46/S45* has lower ratio than *S54/S53*.

Reviewer #2 (Remarks to the Author):

Li and coworkers present a chemoselective strategy between nucleophiles and dienes by using different 3-substituted arynes. In the presence of 3-silyl substituted arynes, cycloaddition reactions occur preferentially over nucleophilic addition reactions, and the employment of 3-methoxybenzyne could switch the chemoselectivity toward nucleophilic addition reactions. The authors examined a variety of functional groups, such as imidazole, N-tosylated/N-alkyl aniline, phenol, alcohol, furan, cyclopentadiene, pyrrole, cycloheptatrienone, and cyclohexene, and found that this strategy could be applied in other types of transformations, such as C–N bond insertion, Truce–Smiles rearrangement, and nucleophilic annulation. Therefore, the substrate scope of this method is broad. Then, the authors give a synthesis of dihydrexidine as a demonstration for the application of their chemoselective strategy. The key steps are a selective 4+2 cycloaddition reaction and the removal of the TBS group, which should inspire people for the potential application of this strategy. At last, they use competing reactions and DFT calculations to understand the origin of the chemoselectivity, which is reasonable.

Although it is known that an electron-withdrawing group on the 3-position of benzyne could control nucleophile to attack its meta position in regioselective manner, interestingly, this study is quite different by controlling the chemoselectivity between nucleophile and dienes. Moreover, it is unexpected that 3-silyl substituted arynes could

be able to pick up dienes from various nucleophiles. This discovery has not been previously revealed. Although there are numerous successful examples on new transformations of benzyne, it is the first time that Li could control chemoselectivity in benzyne reactions. Overall, this reviewer thinks that the study by Li and coworkers is an inspiring and deep investigation. Both the manuscript and the supporting information are well prepared. This reviewer believes that this work is an interesting discovery in benzyne chemistry and meets the requirement for novelty by Nat. Commun. and should be accepted after minor revision.

Reply: *we thank this referee for these comments.*

Questions:

1) Have the authors tried to examine other types of cycloaddition reactions, such as 3+2 and 2+2 cycloaddition reactions?

Reply: *in the course of our study, we have conducted experiments using various [3+2]/[2+2] cycloaddition partners together with nucleophiles and dienes. Both nitron and benzyl azide as [3+2] cycloaddition partners were found to be too reactive that would not serve as a proper partner with furan. On the other hand, silyl enol ether, as a formal [2+2] cycloaddition partner, was found to be inert to couple with furan in our study. So far, we have not been able identify a proper reaction partner with dienes.*

2) Could 3-silyl substituted arynes be used to select a nucleophile among several different nucleophiles? This would be interesting.

Reply: *thanks for this suggestion. Unfortunately, we examined some nucleophiles and found no effect to differentiate the inherent reactivity sequence.*

3) Could the authors explain why 3-substituted arynes other than those reported in Fig. 3b are not applicable in this study?

Reply: *Although 3-methoxybenzyne was found to be the most efficient one in our study, we did find several others and showed them in Fig 3b. However, other 3-substituted benzyne, such as those with Br, Cl, F, OAc, etc., could not give satisfied yields. In fact, those aryne intermediates have generally found to give low reaction yields in many typical reactions. Therefore, our study follows general reactivity trend with respect to those aryne precursors in nucleophilic-type reactions.*

4) As an ultimate goal, it would be great to see chemoselectivity by using simple benzyne. Would it be possible to use unsubstituted benzyne to realize chemoselectivity between nucleophile and dienes?

Reply: *this is a good suggestion. At this moment, we cannot tell how far we should wait until simple benzyne could realize chemoselective reactions. My hypothesis is that it would be plausible that under certain circumstances, such as confined environment, people might be able to use simple benzyne to do this job.*

5) Figure 2A, 15a was derived from a primary alcohol, how about the reaction results when secondary alcohols and tertiary alcohols were utilized?

Reply: *alcohol is not a good nucleophile in benzyne chemistry. In our study, we found that the reactivity of primary alcohol matches well with Alder-ene reaction to produce*

15a selectively. However, in the study on sub-16 (Fig 4) only primary alcohol could match the reactivity with diene, while tertiary alcohol is inert with respect to both 3-methoxybenzyne and 3-silylbenzyne.

6) This work is involved in arene amination, some related work should be cited in a suitable position (J. Am. Chem. Soc. 2017, 139, 15656; J. Am. Chem. Soc. 2018, 140, 4503; Sci. Chin. Chem. 2024, 67, 374; ACS Catal. 2020, 10, 8402; Angew. Chem. Int. Ed. 2019, 58, 10245).

Reply: we added those references to our new manuscript. Thanks.

Reviewer #3 (Remarks to the Author):

In this work, Li and coworkers reported a substrate-controlled switchable chemoselective aryne reaction between nucleophiles and pericyclic arynophiles. Authors developed manipulate chemoselective control methods between two major types of aryne transformations. Nucleophilic addition-triggered reactions and non-polar pericyclic reactions could be differentiated by employing different substituted arynes. This substrate controlled chemoselective protocol was found to be applicable between various nucleophiles, the applications showed that the method had a good perspective in the field of synthetic application. Control experiments were designed and carried out carefully. The manuscript was well organized, and should be interesting to the general readership in the field of synthetic chemistry. Thus, I agree to the publication of this manuscript in Nature Communications after addressing the following minor points.

Reply: we thank this referee for these comments.

1. As for the subsequent research results corresponding to the conditional screening given in Fig. 1c, most insertion reactions require the participation of 18-c-6. But when 3-(TBS) benzene precursors occur [4+2] cycloaddition reactions, 18-c-6 is hardly required. However, the compounds 20b, 21b, 22b and sub-15 all required the participation of 18-c-6. Did it conflict with each other? What role did 18-c-6 play in the reaction?

Reply: all benzyne reactions are not single-step transformations. They all include a benzyne generation step. Due to the high reactivity and short lifetime of a benzyne intermediate, slow releasing of benzyne from its precursor in a reaction media is necessary. Therefore, different reaction conditions including various solvents, fluoride salts, and/or 18-c-6 are normally used in order to tune the generation rate of benzyne so that it can match well with the consequent reactions. Although the role of 18-c-6 is to increase the solubility of the fluoride salt, this solubility should maintain certain low level to assure the high efficiency of the following reaction. In our study, we examined different types of reaction modes with quite different substrates. For each substrate, we had to screen different reaction conditions so that it could reach the optimal selectivity and yield. Due to the reactivity of different nucleophile/dienes, benzyne generation rate should be different. To this end, in some cases we found 18-c-6 is necessary, but in other cases it could not give satisfied results.

2. When the reaction of 3-substituted aromatic hydrocarbons to nucleophilic and

pericyclic aromatic hydrocarbons were controlled chemically, did the choice of reaction solvent affect the selectivity of the reaction? If so, can you summarize the general pattern? How did different types of bases affect the reaction?

Reply: *as mentioned in the previous question, maintaining a suitable low concentration of fluoride in the reaction system would allow slow releasing of benzyne. Therefore, it could match well with the following chemoselective reactions. A proper combination of reaction solvent, fluoride salt, 18-c-6, and temperature is the key for the success of a benzyne reaction. For each substrate, we had to screen several reaction conditions so that it could reach the optimal selectivity and yield. Unfortunately, there is no clear trend or general pattern in our study because different reactions modes were included. For a typical benzyne reaction, CsF/MeCN and KF/18-c-6/THF are the two common reaction conditions, but in many cases they are not applicable and the optimal conditions should be case by case. Therefore, there is no definite reaction conditions that could cover different types of substrates in our study. To response this question by this reviewer, we added general procedures for the reactions with those aryne precursors to the end of our manuscript, where they could be applied to most of the substrates. As for the role of bases in the reactions of **sub-1**, **sub-6**, and **sub-7** (Fig 2) with 3-methoxybenzyne, the bases could deprotonate those nucleophile so as to increase their nucleophilicity.*

REVIEWERS' COMMENTS

Reviewer #1 (Remarks to the Author):

In the revised manuscript, the authors have adequately addressed all of my comments on the original paper and the significance of their work has been clarified. This reviewer considers this paper has become publishable in Nature Communications. The following minor points are recommended to be corrected prior to publication, e.g., during the galley proof phase.

1. Concerning the comment #1-1, it is recommended to define or change the term “pericyclic arynophile,” also used in the title, as this is not commonly used.
2. Concerning the comment #1-2, the modification to the reaction conditions of sub-6 and sub-7 should be mentioned in the main text.
3. Concerning the comment #1-5b, the logic for determining the stereochemistry of S45/S46, S49/S50 appears reasonable, but it would be helpful to note that these are estimated structures.

Reviewer #1 (Remarks to the Author):

In the revised manuscript, the authors have adequately addressed all of my comments on the original paper and the significance of their work has been clarified. This reviewer considers this paper has become publishable in Nature Communications. The following minor points are recommended to be corrected prior to publication, e.g., during the galley proof phase.

Reply: we thank this referee for these comments.

1. Concerning the comment #1-1, it is recommended to define or change the term “pericyclic arynophile,” also used in the title, as this is not commonly used.

Reply: thanks for this suggestion. We changed “pericyclic arynophile” to “pericyclic reaction partner”.

2. Concerning the comment #1-2, the modification to the reaction conditions of sub-6 and sub-7 should be mentioned in the main text.

Reply: we changed “To this end, 12a and 13a were obtained from sub-6 and sub-7 in good yields after arylation with 1b (Fig. 2a)” to “To this end, 12a and 13a were obtained from sub-6 and sub-7 in good yields after arylation with 1b in the presence of carbonate salts (Fig. 2a)”.

3. Concerning the comment #1-5b, the logic for determining the stereochemistry of S45/S46, S49/S50 appears reasonable, but it would be helpful to note that these are estimated structures.

Reply: we added notes at the corresponding positions in the Supplementary Information section. Thanks for this suggestion.